# The Mechanisms Underlying the Beneficial Impact of Aerobic Training on Cancer-Related Fatigue: A Conceptual Review

**DOI:** 10.3390/cancers16050990

**Published:** 2024-02-29

**Authors:** Adeline Fontvieille, Hugo Parent-Roberge, Tamás Fülöp, Michel Pavic, Eléonor Riesco

**Affiliations:** 1Faculty of Physical Activity Sciences, University of Sherbrooke, 2500 Boulevard de l’Université, Sherbrooke, QC J1K 2R1, Canada; adeline.fontvieille@usherbrooke.ca (A.F.); hugo.parent-roberge@usherbrooke.ca (H.P.-R.); 2Research Centre on Aging, 1036 Rue Belvédère Sud, Sherbrooke, QC J1H 4C4, Canada; tamas.fulop@usherbrooke.ca; 3Institut de Recherche sur le Cancer de l’Université de Sherbrooke, 12e Avenue N Porte 6, Sherbrooke, QC J1H 5N4, Canada; michel.pavic@usherbrooke.ca; 4Faculty of Medicine and Health Sciences, University of Sherbrooke, 3001 12e Avenue N, Sherbrooke, QC J1H 5N4, Canada

**Keywords:** immune system, hypothalamic–pituitary–adrenal axis, inflammation, neuroinflammation, aerobic exercise, acute exercise

## Abstract

**Simple Summary:**

Cancer-related fatigue is a prevalent symptom, with a significant impact on the daily lives of those affected. While physical exercise has demonstrated effectiveness in reducing the intensity and duration of fatigue, the literature still lacks sufficient evidence on the physiological mechanisms explaining this impact. This conceptual review aimed to provide an overview of the evidence regarding the effect of acute exercise on peripheral and neuronal inflammation, immune function, and the neuroendocrine system in the context of cancer. We aim to integrate these pathways into a conceptual model that can serve as a starting point for further research into the physiological mechanisms linking exercise and cancer-related fatigue.

**Abstract:**

Cancer-related fatigue (CRF) is a prevalent and persistent issue affecting cancer patients, with a broad impact on their quality of life even years after treatment completion. The precise mechanisms underlying CRF remain elusive, yet its multifaceted nature involves emotional, physical, and cognitive dimensions. The absence of effective medical treatments has prompted researchers to explore integrative models for potential insights. Notably, physical exercise emerges as a promising strategy for managing CRF and related symptoms, as studies showed a reduction in CRF ranging from 19% to 40%. Current recommendations highlight aerobic training at moderate intensity as beneficial, although questions about a dose–response relationship and the importance of exercise intensity persist. Despite the positive impact of exercise on CRF, the underlying mechanisms remain elusive. This review aims to provide a theoretical model explaining how aerobic exercise may alleviate CRF. Focusing on acute exercise effects, this review delves into the potential influence on peripheral and neural inflammation, immune function dysregulation, and neuroendocrine system disruptions. The objective is to enhance our understanding of the intricate relationship between exercise and CRF, ultimately paving the way for tailored interventions and potential pharmacological treatments for individuals unable to engage in physical exercise.

## 1. Introduction

Cancer-related fatigue (CRF) is a significant burden for patients [1,2], affecting 14% to 99% of them, depending on various factors such as cancer stage, treatment type, and assessment methods [3,4]. While recognized as a multidimensional syndrome encompassing emotional, physical, and cognitive manifestations [5], CRF is defined by the National Comprehensive Cancer Network as “an unusual, persistent, subjective sense of tiredness related to cancer or cancer treatment that interferes with usual functioning” [6]. CRF can persist more than 10 years after the end of cancer treatments [7,8] and has a deleterious impact on health-related quality of life [9]. Usually, this fatigue sets in within a few days of treatment, peaking 1 to 3 days post-chemotherapy, and then gradually subsides, returning to values near those before treatment around 10 days post-chemotherapy (as shown in Figure 1). There is currently no effective medical treatment for CRF, mainly because its underlying mechanisms remain unclear. In the past decade, several groups have attempted to fill this gap by suggesting integrative models that consider potential mechanisms involved in CRF [10,11,12,13,14]. Despite the lack of established medical treatments, these models and the systemic impact of physical exercise suggest exercise as a promising and effective strategy for managing CRF [15] and other treatment-related symptoms [16]. In fact, while we observed a reduction in CRF for up to 19% in oncogeriatric patients with exercise training [17], others have reported a greater impact in adjuvant breast cancer with a 40% reduction [16]. Currently, it is suggested that aerobic training at moderate intensity (65% heart rate max [HRmax] or 45% VO_2_ max) reduced CRF, while it remains unclear if there is a dose–response relationship and if exercise intensity matters [18,19,20]. Nevertheless, there is a paucity of data regarding the underlying mechanisms by which exercise training reduces CRF. Unveiling the mechanisms by which exercise reduces CRF is crucial for developing personalized interventions and potential pharmacological alternatives for individuals unable to exercise. Given that chronic exercise is the outcome of accumulating acute exercise effects [21], we can infer that the decrease observed after an exercise intervention may be explained by the repetitive impact of acute exercise effects on CRF (Figure 1).

While there is no accepted mechanistic model explaining the reduction in CRF induced by exercise, it would be possible to provide such a theoretical model by using available evidence from exercise studies performed with cancer patients, or other chronic diseases like chronic fatigue syndrome [22], as well as healthy patients. In fact, among the suggested mechanisms involved in CRF etiology, some of them may be acutely influenced by aerobic exercise, such as peripheral pro-inflammatory state [23], immune function dysregulation [14,24], as well as neuroendocrine dysregulations [25,26]. Hence, the objective of this conceptual review is to provide a theoretical model explaining how aerobic exercise may reduce CRF. Therefore, this review will mostly focus on the acute effect of exercise on peripheral and neural inflammation, as well as dysregulation of the immune function and neuroendocrine system. Nevertheless, a brief review of the current literature regarding potential mechanisms explaining CRF precedes the content section to facilitate understanding (for a more detailed review, please see [10,11,27]).

## 2. The Effects of Peripheral and Neural Inflammation on Cancer-Related Fatigue

### 2.1. Effect of Chemotherapeutic Treatments

Chemotherapy treatments, primarily used for cancer, trigger an acute response from both the innate and adaptive anti-tumor immune systems [28]. However, prolonged treatment and the accumulation of chemotherapeutic agents can lead to changes in the immune system within the tumor environment, including tumor cell death, and damage to healthy tissues due to chemicals. In response to this acute disturbance in the body’s homeostatic, the immune system initiates a protective response by synthesizing cytokines. The objective is to reduce tissue damage by increasing the production of pro-inflammatory cytokines and chemokines, which help stimulate and release lymphocytes and monocytes in the affected tissue. This acute inflammatory response leads to an increase in both pro- and anti-inflammatory cytokines. However, with repeated chemotherapy treatments, this response becomes detrimental, leading to a maladaptive chronic inflammatory state. Studies have shown an increase in various inflammatory markers in the peripheral circulation during cancer [29], including IL-1, IL-6, IL-1β, and TNF-α in different types of cancer [30,31,32]. Notably, this rise of inflammatory cytokines coincides with a peak in CRF levels, as shown by Raudonis et al. [33].

Progressively, the repeated elevation of pro-inflammatory cytokines after each treatment cycle triggers a rise in circulating anti-inflammatory cytokines [34,35], ultimately contributing to the maladaptive chronic inflammation mentioned above. This peripheral inflammation disrupts the blood–brain barrier (BBB) and activates the microglia, the resident macrophage population responsible for immune defense in the central nervous system (CNS) [36]. Repeated microglial activation initiates a harmful cycle, leading to the production of more pro-inflammatory cytokines and the activation of neurotoxic reactive astrocytes—both contributing to neuroinflammation [37]. Additionally, the compromised BBB allows for increased permeability [37]. It was previously shown that chemotherapeutic substances can reach and cross the BBB due to a negative impact of some pro-inflammatory cytokines (IL-1α and IL-1β) on tight junctions [38,39], and both IFN-γ and TNF-α can alter BBB permeability by affecting the expression and cellular distribution of junctional adhesion molecules (i.e., ICAM-1 and VCAM-1) [40]. This disruption enables an elevated number of blood-derived molecules and cells, such as activated T cells and B-cells, to enter the CNS and initiate neurodegeneration that could promote CRF. Microglia activation can also initiate an inflammatory cascade that eventually alters the metabolic pathways within CNS such as the kynurenine pathway, ultimately leading to impairments in the releasing of neurotransmitters such as serotonin and especially dopamine [41,42]. It is well established that several chemotherapeutics agents can negatively impact the hippocampal region, leading to a decreased cell proliferation or increased cell death [43]. This effect can manifest as behavioral changes, including cognitive, emotional, and spatial impairment in cancer patients. Dopamine signaling plays a crucial role in influencing motivational states [44], which, in turn, can impact hippocampal function. Dysfunction in any of these components can potentially lead to alterations in motivational behaviors and cognitive processes.

### 2.2. The Acute Effect of Aerobic Exercise on Peripheral and Neural Inflammation

Aerobic exercise can have a beneficial acute effect on the peripheral inflammatory profile. During exercise, a rise in pro-inflammatory cytokines occurs while a switch takes place after the end of exercise, with an increase in circulating anti-inflammatory cytokines [45]. More precisely, according to the intensity [46] and duration of aerobic exercise, many inflammatory myokines increase, especially IL-6 [47]. In response to a short bout of aerobic exercise, it was proposed that the release of myokine IL-6 could upregulate the production of anti-inflammatory cytokines, including IL-10, IL-1Ra, and TGF-β, which would dampen the inflammatory response for several hours after exercise [48,49]. When these anti-inflammatory cytokines are carried into the central nervous system (CNS), they might help reduce neural inflammation [50], potentially leading to changes in behavior. Indeed, in a large meta-analysis of 18 studies [51], evidence showed that acute moderate-intensity exercise led to an increase in anti-inflammatory cytokines, with a marked response observed after high-intensity exercise. However, it is not clear whether this increase in post-exercise anti-inflammatory cytokines leads to a decrease in CRF, as no study has directly measured this relationship. Therefore, because high peripheral levels of anti-inflammatory cytokines inhibit microglial activation by interacting with brain cytokine receptors [52], it may be possible that during the following hours after aerobic exercise, the neural activation is reduced, which could explain the lower CRF observed in response to exercise.

The BBB also benefits from aerobic exercise, notably by restoring permeability and re-establishing the expression of tight junction transmembrane proteins [53]. This effect was observed in a mouse model of multiple sclerosis, where endurance exercise led to a significant decrease in IFN-γ and IL-1β production within the CNS, contributing to the protection of the BBB [53]. These findings imply that aerobic exercise can help restore the function of the BBB, which, in turn, reduces inflammation in the CNS. This prevents the entry of neurotoxic metabolites from the kynurenine pathway into the CNS, potentially resulting in a decrease in CRF.

## 3. Immune Function Dysregulation and Cancer-Related Fatigue

### 3.1. Immune Response after Chemotherapy

While the precise role of immune cells in the development of CRF is not yet fully understood, recent evidence indicates that immune system dysregulation, including the activity of natural killer (NK) cells and T lymphocytes, might be involved in the onset and persistence of CRF. T cells, specifically CD4^+^ and CD8^+^, are key players in adaptive immunity and have direct antitumor effects [54,55]. CD4^+^ T cells, also known as T helper, help coordinate immune responses by releasing cytokines and activating other immune cells. CD8^+^ T cells, also called cytotoxic T cells, directly kill infected or cancerous cells. Dysregulation of T lymphocytes, including a shift in the balance between different T cell subsets and altered cytokine production, has been observed in myalgic encephalomyelitis/chronic fatigue syndrome (ME/CFS) patients [56]. These dysfunctions can result in impaired immune responses against cancer cells, which can contribute to disease progression and fatigue. However, there is currently no evidence linking T-cell dysfunction in cancer patients to CRF. Moreover, research on ME/CFS patients demonstrated the reduced activity of NK cells [57,58], which has also been reported in cancer patients experiencing fatigue. Two distinct populations of NK cells can be found in peripheral circulation based on their cell surface density, the CD56^bright^ (immature) and CD56^dim^ (cytotoxic). A recent study showed a slight but significant decrease in the population of cytotoxic NK cells (CD56^dim^) in patients with CRF compared to a non-fatigued group [24], suggesting a potential involvement of impaired NK cell function in CRF. However, the reason for reduced NK cell cytotoxic activity in cancer patients with CRF is yet to be determined.

Altogether, these dysregulations can promote the release of pro-inflammatory cytokines and other signaling molecules, which then contribute to a state of chronic inflammation. The persistent activation of the immune system and the resulting inflammatory response can induce CRF by affecting the CNS, altering neurotransmitter levels, and disrupting normal energy metabolism.

### 3.2. Acute Effect of Aerobic Exercise on Immune Markers

Growing evidence suggests that aerobic exercise may improve CRF by modulating the immune system. Research has shown that exercise improves immunosurveillance by increasing the number of lymphocytes in the peripheral circulation during exercise in healthy adults [59], a key component of the immune response, although this effect diminishes after cessation. While we recently finished a proof-of-concept study investigating the relationship between acute exercise-induced immune response and CRF (clinicalTrials.gov ID: NCT04715061), it was previously reported that a single bout of moderate-intensity aerobic exercise is sufficient to enhance the mobilization of NK cells and recruitment of cytotoxic T cells (CD8^+^) [60], with both implicated in immune surveillance. Since NK cell [61] and T cell [62] dysfunction has been linked to fatigue in various populations, it may be possible that exercise-induced improvements in immune function contribute to reduced CRF.

Supporting this notion, studies by Campbell et al. [60] and others [63,64] have demonstrated that high-intensity aerobic exercise (85% peak power output) elicits a transient greater lymphocyte mobilization in the peripheral circulation compared to moderate-intensity in healthy individuals, then returning to baseline values after 60 min post-exercise. Moreover, our preliminary results (under review) further suggests that aerobic exercise, particularly high-intensity exercise in fitter individuals, can promote acute lymphocyte mobilization in the peripheral circulation in metastatic cancer patients [65]. Additionally, high-intensity exercise has been shown to induce a more pronounced increase in NK cell cytotoxic activity compared to light- or moderate-intensity aerobic exercise [66,67]. This transient mobilization of cytotoxic NK cells in peripheral blood might be transported into the tumor microenvironment. However, further research is necessary to comprehensively characterize the extent of immune modulation induced by different aerobic exercise intensities in cancer patients and to definitively establish the link between these changes and CRF.

## 4. Neuroendocrine Alteration and Cancer-Related Fatigue

### 4.1. The Influence of Chemotherapeutic Agents on the Neuroendocrine System

Neuroendocrine alterations may also contribute to CRF as reviewed by O’Higgins et al. [27]. In fact, chronic inflammation tends to reduce the synthesis and release of corticotropin-releasing hormone (CRH) [27,68], which is a central regulator of the hypothalamic–pituitary–adrenal (HPA) axis. This disrupts the HPA axis, which negatively impacts the regulation of stress hormone cortisol’s synthesis and release. Furthermore, studies have observed either resistance or sensitivity to glucocorticoids [69], suggesting a potential disruption in the negative feedback loop regulating CR, ACTH, and cortisol levels. This impaired regulation may lead to the HPA system releasing cortisol at a constant level throughout the day in cancer patients. Supporting this notion, a study conducted by Bower et al. [70] reported higher daytime cortisol levels with a blunted circadian rhythm (flatter slope) compared to normal peak times (morning and evening), suggesting dysfunction of the HPA axis in patients with CRF. However, the relationship between HPA dysfunction and CRF is still controversial. It is worth noting that some cancer treatments, such as glucocorticoids and certain chemotherapy drugs, can affect the HPA axis, potentially contributing to CRF. However, this appears to be more common when glucocorticoid withdrawal coincides with adrenal insufficiency [71].

Interestingly, indoleamine-2,3-dioxygenase (IDO) is a counter-regulatory enzyme that contributes to immune suppression in the tumor microenvironment [72]. By numerous mechanisms reviewed in Johnson et al. [73], greater activation of IDO expression stimulates tryptophan catabolism, resulting in increased circulating levels of kynurenine (KYN) to the detriment of serotonin [74]. Knowing that serotonin regulates upstream CRH signaling systems [75], a decrease in serotonin levels might reduce the activity of the HPA axis and impair cortisol production [27,76]. Studies conducted in cancer patients have shown a correlation between IDO activity and CRF severity, indicating that a low tryptophan concentration or a high ratio of kynurenine/tryptophan was correlated with high CRF or lethargic behaviors. IDO activity is closely linked to the immune system, as the peripheral mononuclear cells have been shown to be potent producers of IDO [77]. In an interesting way, KYN promotes the generation of T_reg_ cells and also inhibits the proliferation of NK cells, B cells, and CD4^+^ and CD8^+^ lymphocytes [78], which can be argued to play a role in the reduction of immune defense. Similarly, the hypothesis of dopaminergic imbalance appears to be akin to serotonin dysregulation. In this context, both low and excessive levels of dopamine can induce fatigue in individuals with multiple sclerosis [42]. In this scenario, T cells, especially CD4^+^ cells, can breach the BBB in the CNS and trigger the production of IFN-γ, which, in turn, inhibits dopamine production and can even lead to the destruction of dopamine neurons. Finally, as the kynurenine metabolites reach the BBB and enter the CNS, they become metabolized into quinolinic acid by glial cells (e.g., microglia) and cause neurotoxicity, potentially leading to CRF.

### 4.2. The Acute Effect of Aerobic Exercise on the Neuroendocrine System

In healthy humans, the acute response to moderate-to-vigorous exercise (>60% of VO_2_ max) stimulates the HPA axis by increasing adrenocorticotropic hormone (ACTH) and cortisol levels [42]. Nevertheless, two studies on cancer survivors revealed that during moderate-intensity exercise (60% VO_2_ peak), the magnitude of the increase in cortisol and ACTH blood concentration is smaller [79,80], mainly because of the higher baseline levels observed in cancer patients compared to healthy individuals. Hence, exercise-induced HPA axis activation may be somewhat limited in cancer survivors. However, to which extent aerobic training performed over several weeks in cancer patients during treatment can help restore normal baseline circulating cortisol and ACTH levels remains to be investigated.

The sensitivity of glucocorticoid receptors, principal inflammatory regulators of the HPA axis, is also affected by aerobic exercise. In fact, it was demonstrated that exercise acutely increased tissue sensitivity to glucocorticoids, which could be explained by an increasing number of glucocorticoid receptors or a shift in their isoform expression [81]. Hence, cortisol has more potential to bind to glucocorticoid receptors, which would restore appropriate negative feedback. It could then be hypothesized that exercise allows for the recovery of diurnal cortisol variations via the enhancement of glucocorticoid receptors sensitivity, thus providing a negative feedback loop.

At the central level, exercise acutely acts on dopaminergic and serotoninergic pathways, which are suggested to be contributors to CRF. Acute exercise increases the bioavailability of free tryptophan in the brain, along with an increase in tryptophan hydroxylase, the enzyme that converts tryptophan into serotonin. By doing this, exercise leads to an increase in serotonin levels in the hours after exercise [82]. Even though there are still discrepancies in the literature regarding the required intensity of exercise to trigger increased cerebral serotonin levels in cancer patients, it is accepted that high-intensity exercise promotes higher serotonin and dopamine levels [83]. An adequate concentration of these two neurotransmitters ensures their optimal function and could consequently decrease CRF.

IDO activation, which was previously associated with a high level of CRF [84,85], could also be influenced by aerobic exercise. Studies investigating the acute effect of aerobic exercise showed an increase in kynurenine concentration in circulation [86,87]. This is mainly explained by the transient elevation of pro-inflammatory cytokines during exercise [88]. However, the acute anti-inflammatory impact of exercise in the following hour could also decrease IDO activity [89] and thus restore normal tryptophan metabolism. Regarding BBB permeability, acute exercise suggests the re-establishment of the BBB function and permeability and might prevent the crossing of neurotoxic metabolites from the kynurenine pathway to the CNS [36]. However, little evidence is available on this phenomenon in acute exercise in cancer patients.

## 5. Physiological Variability in Exercise Response

Inter-individual variability in physiological responses following exercise is observed in both healthy populations and cancer patients [90] and is influenced by a multitude of factors such as genetics, baseline fitness levels, age, and demographic variables like sex and ethnicity. Studies have demonstrated a wide range of responses to exercise, with individuals experiencing either increased fatigue or improvements in factors like cardiorespiratory fitness. Among the explaining factors, a genetic component likely contributes, as twin and family studies revealed a genetic component explaining 30 to 60% of the variation in cardiorespiratory fitness response to exercise [91]. Distinct responses observed between “low” and “high” responders in animal experiments provide evidence for genetic factors influencing training adaptations and thus possibly the impact of a single exercise session. Additionally, the type, intensity, duration, and timing of exercise play a significant role in shaping the outcome. Inconsistency in exercise prescription across studies hinders the comparison of true variability in response to exercise. Studies have employed various approaches for standardization, including fixed duration, intensity, or caloric expenditure targets. Most of the existing literature fails to report exercise interventions following CERT guidelines, often only considering adherence, defined as exercise session attendance, solely. Moreover, monitoring methods like heart rate can be affected by cardiovascular drift and pharmacological treatment, potentially leading to underestimating the actual workload. Beyond these factors, behavioral aspects like overall physical activity levels, dietary habits, and sleep quality contribute to the variability observed in exercise responses. This is particularly relevant for cancer patients due to their unique characteristics, including pre-existing comorbidities, fitness levels, and specific cancer diagnoses (e.g., initial cancer type, the presence and location of metastases, and treatment regimen).

Overall, the hypothetical mechanisms by which aerobic exercise can decrease CRF are presented in Figure 2. Keeping in mind that this is a proposed theoretical model, the scientific literature on chronic fatigue syndrome provides a good starting point for future studies.

## 6. Conclusions

Although the underlying mechanisms explaining how aerobic exercise impacts CRF are still under investigation, this conceptual review proposes an explanatory theoretical model of exercise based on the suggested etiology of CRF. While this review focuses on the physiological aspects and immediate impacts of chemotherapy treatment, shedding light on these aspects, the etiology of CRF is intricate and involves multiple factors, such as type of treatment, type of cancer, and chemotherapeutic agents. Additional factors mentioned in the literature may account for other aspects of CRF, such as physical deconditioning, depression, and cachexia. Understanding these biological mechanisms is crucial for developing personalized exercise interventions for cancer patients and mitigating the impact of this significant side effect on their quality of life. Further studies and a proof of concept are required to confirm or refute this model and assess its applicability based on the specifics of cancer treatment, such as the type and drugs used. Moreover, knowing that aerobic exercise is currently one of the most efficient strategies to reduce CRF, a better understanding of the underlying mechanisms would offer the opportunity to develop pharmacological treatments for individuals who cannot exercise or achieve a sufficient exercise workload to benefit from this type of intervention. By enhancing our knowledge of these mechanisms, more precise exercise prescriptions could help patients increase their survival rate after diagnosis or reduce the risk of recurrence [92], as well as alleviate treatment-related side effects such as CRF [93], initiating a virtuous circle. Establishing recommendations based on new evidence regarding the comprehension of CRF mechanisms will provide important leverage in implementing exercise as an integral part of the healthcare pathway for individuals with cancer.

## Figures and Tables

**Figure 1 cancers-16-00990-f001:**
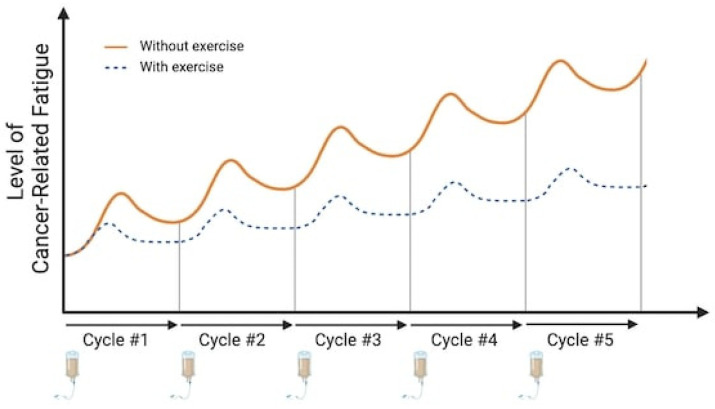
Kinetics of cancer-related fatigue depending on the cycle of chemotherapy treatment and exercise. Figure 1 illustrates the kinetics of CRF in relation to chemotherapy treatment cycles and their cumulative effect. The orange curve depicts a characteristic pattern of CRF intensity following each treatment cycle. A peak in CRF is observed between 3 and 5 days post-treatment, followed by a recovery period in the subsequent days. Notably, the intensity of CRF progressively increases with the accumulation of treatments. The blue curve depicts the fatigue kinetics when physical exercise is incorporated during treatment cycles, highlighting the potential for chronic effects. Here, a marked reduction in CRF intensity is observed compared to the no-exercise scenario. Additionally, the cumulative anti-cancer treatment effect appears less pronounced with exercise intervention. This observation suggests that the acute effect of exercise, practiced within a treatment cycle, might contribute to a gradual attenuation of CRF over time, potentially reflecting the cumulative impact of acute exercise. Image created with Biorender.com (accessed on 6 December 2023).

**Figure 2 cancers-16-00990-f002:**
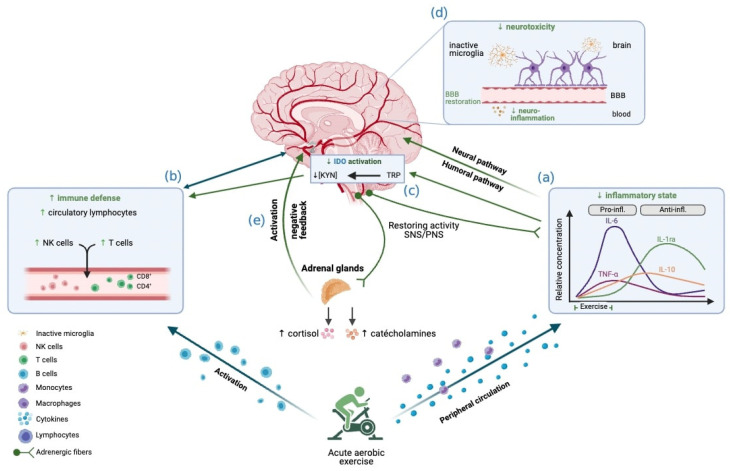
Effects of acute aerobic exercise on CRF, integrated into the conceptual model. The reduction in CRF observed following aerobic exercise may be explained by these mechanisms: Aerobic exercise, particularly when performed at high intensity, is associated with an anti-inflammatory effect during the first few hours after exercise, possibly transiently alleviating chronic peripheral inflammation (a). Aerobic exercise promotes the mobilization of circulating lymphocytes, thus helping to restore immune function (b). This reduction in inflammation would lead to a reduction in the activation of IDO (c), with effects accentuated by vigorous intensity. The restoration of BBB function prevents the passage of pro-inflammatory cytokines into the CNS, which decreases microglia activation and therefore prevents neurotoxic damage (d). Aerobic exercise manages to re-regulate the activation of the HPA axis (e), notably by recovering the amplitude of cortisol release, while promoting a better sensitivity to glucocorticoids and restoring the negative feedback loop. Image created with Biorender.com (accessed on 22 September 2022).

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
