# Peer review of "The Mechanisms Underlying the Beneficial Impact of Aerobic Training on Cancer-Related Fatigue: A Conceptual Review"

_cancers, 2024, doi:10.3390/cancers16050990_

Round 1

Reviewer 1 Report

Comments and Suggestions for Authors

General comments:

1. The paper presents a literature review on potential mechanisms underlying the beneficial impact of aerobic training on cancer-related fatigue .

2. The manuscript is well-written and understandable.

3. The research topic is of relevance for research, but also for medical practice.

Specific comments:

4. The paper focuses on physiological mechanisms underlying the beneficial impact of aerobic training on cancer-related fatigue . However, motivational and affective mechanisms seem also important.

5. In general, the underlying mechanisms could be better specified in more detail to further elaborate the conceptual claims.

6. Inter-individual differences could be considered in more depth. Some individuals may function differently than other patients.

7. A lot of confounding factors may be at work that could be linked to the variables of interest. Those should be discussed further.

8. The authors may specify in more detail how the PRISMA guidelines were followed.

9. Some results across the inspected studies may be integrated using meta-analytic techniques.

10. The practical implications could be illustrated with a more detailed view on the participants’ lives.

Author Response

Comments 1: The paper presents a literature review on potential mechanisms underlying the beneficial impact of aerobic training on cancer-related fatigue, (1) the manuscript is well-written and understandable and (2) the research topic is of relevance for research, but also for medical practice.

Response 1: Thank you for your thoughtful feedback on the article, particularly your appreciation for its relevance and clear organization of concepts.

Comments 2: The paper focuses on physiological mechanisms underlying the beneficial impact of aerobic training on cancer-related fatigue. However, motivational and affective mechanisms seem also important.

Response 2: Thank you for raising this crucial point. We acknowledge that various factors, including motivational and psychological aspects, likely contribute cancer-related fatigue to varying degrees. While our primary focus in this study is on the biological components to gain a deeper understanding of this phenomenon, we recognize the potential influence of other dimensions as you mentioned. To address this, we have incorporated a brief discussion on dopaminergic information within the section " 2.1. Effect of chemotherapeutic treatments”. While acknowledging the limited research on the direct impact of acute exercise on motivational aspects in cancer patients, we highlight the potential connection between dopamine signaling, motivational states, and hippocampal function.

The following section is now added (Page 4, Lines 148-155):

 “It is well-established that several chemotherapeutics agents can negatively impact the hippocampal region, leading to a decreased cell proliferation or increased cell death [42]. This effect can manifest as behavioral changes, including cognitive, emotional, and spatial impairments, in cancer patients. Dopamine signaling plays a crucial role in influencing motivational states [43], which in turn can impact hippocampal function. Dysfunction in any of these components can potentially lead to alterations in motivational behaviors and cognitive processes.”

Comments 3: In general, the underlying mechanisms could be better specified in more detail to further elaborate the conceptual claims.

Response 3: We appreciate the reviewer's comment and have addressed the request for a more detailed explanation of the underlying mechanisms by adding specific information in the following sections:

- We have incorporated details on how chemotherapeutic agents can disrupt the blood-brain barrier (BBB) through the negative impact of pro-inflammatory cytokines and the alteration of junctional adhesion molecules by IFN-γ and TNF-α.

We added the following information:

Page 4, Line 139: “It was previously shown that chemotherapeutic substances can reach and cross the BBB due to the disruption of tight junctions by some pro-inflammatory cytokines (IL-1α and IL-1β) [38,39]. Additionally, both IFN-γ and TNF-α can alter BBB permeability by affecting the expression and cellular distribution of junctional adhesion molecules, such as ICAM-1 and VCAM-1 [40].”.

We added the following details about the role of resistance glucocorticoids into the negative feedback loop of HPA axis:

-Page 6, Line 249: “Furthermore, studies have observed either resistance or sensitivity to glucocorticoids [69], suggesting a potential disruption in the negative feedback loop regulating CRH, ACTH, and cortisol levels. This impaired regulation may lead to the HPA system releasing cortisol at a constant level throughout the day in cancer patients. Supporting this notion, a study by Bower et al. [70] reported higher daytime cortisol levels with a blunted circadian rhythm (flatter slope) compared to normal peak times (morning and evening)”.

We added the following details about the link between IDO and immune system, and also some precision about the BBB permeability :

-Page 6, Line 271: “IDO activity is closely linked to the immune system, with peripheral mononuclear cells identified as potent producers of IDO [77]. Interestingly, KYN promotes the generation of Treg cells while inhibiting the proliferation of NK cells, B cells, and CD4+ and CD8+ lymphocytes [78], contributing to reduced immune function.”.

-Page 7, Line 317: “Regarding the BBB permeability, acute exercise suggests an improvement in BBB function and permeability, potentially preventing the passage of neurotoxic metabolites from the kynurenine pathway to the CNS [36]. However, limited evidence is available to confirm this phenomenon in cancer patients with acute exercise.”.

Comments 4: Inter-individual differences could be considered in more depth. Some individuals may function differently than other patients.

Response 4: We appreciate your highlighting the inter-individual variability in response to exercise. To address this, we have incorporated a paragraph acknowledging the potential limitations in the generalizability of this model across diverse cancer types and treatment regimens. However, we opted against delving deeper into this topic due to the inherent challenges in comprehensively capturing the nuances of individual responses within the current scope of the conceptual model, which already relies on certain assumptions.

A section was added in page 7, Line 321:

“Inter-individual variability in physiological responses following exercise is observed in both healthy populations and cancer patients [90] and influenced by a multitude of factors such as genetics, baseline fitness levels, age, and demographic variables like sex and ethnicity. Studies demonstrate a wide range of responses to exercise, with individuals experiencing either increased fatigue or improvements in factors like cardiorespiratory fitness. Among the explaining factors, genetic component likely contributes as twin and family studies revealed a genetic component explaining 30-60% of the variation in cardiorespiratory fitness response to exercise [91]. Distinct responses observed between "low" and "high" responders in animal experiments provide evidence for genetic factors influencing training adaptations. Additionally, the type, intensity, duration, and timing of exercise play significant roles in shaping the outcome. Inconsistency in exercise prescription across studies hinders the comparison of true variability in response to exercise. Studies employ various approaches for standardization, including fixed duration, intensity, or caloric expenditure targets. Most existing literature fails to report exercise interventions following CERT guidelines, often considering only adherence, defined solely as exercise session attendance. Moreover, monitoring methods like heart rate can be affected by cardiovascular drift and pharmacological treatment, potentially leading to underestimating actual workload. Beyond these factors, behavioral aspects like overall physical activity levels, dietary habits, and sleep quality contribute to the variability observed in exercise responses. This is particularly relevant for cancer patients due to their unique characteristics, including pre-existing comorbidities, fitness levels, and specific cancer diagnoses (e.g., initial cancer type, presence and location of metastases, treatment regimen).”

Comments 5: A lot of confounding factors may be at work that could be linked to the variables of interest. Those should be discussed further.

Response 5: We appreciate the reviewer's comment and we have addressed this by incorporating this point within the previous changes. As we thought that some of the inter-individual variation might be interconnected with the confounding factors mentioned, we had elaborated a more global response to these two aspects.

Comments 6: The authors may specify in more detail how the PRISMA guidelines were followed.

Response 6: Thank you for pointing this out. According to the 'Instructions for Authors' of Cancers, the PRISMA guidelines are required for systematic reviews and meta-analyses only. Since our submission is intended as a conceptual narrative review, it is not necessary to include this information in the article.

Comments 7: Some results across the inspected studies may be integrated using meta-analytic techniques.

Response 7: Thank you for your comment. However, as we mentioned above, we have not carried out a methodological analysis since this is a narrative conceptual review.

Comments 8: The practical implications could be illustrated with a more detailed view on the participants’ lives.

Response 8: Thank you for your remarks regarding the participants' point of view. We believe that a better understanding of these CRF mechanisms can help improve targeted exercise prescriptions and optimize the beneficial effects of exercise on cancer-related fatigue. Hopefully, this reduction in CRF may initiate a virtuous circle to regain control over quality of life and improve physical, psychological, and social health.

In line with this, we have added the following paragraph to the conclusion section on page 9, Line 381: 'By enhancing our knowledge of these mechanisms, more precise exercise prescriptions could help patients increase their survival rate after diagnosis or reduce the risk of recurrence [92], as well as alleviate treatment-related side effects such as CRF [93], initiating a virtuous circle. Establishing recommendations based on new evidence regarding the comprehension of CRF mechanisms will provide significant leverage in implementing exercise as an integral part of the healthcare pathway for individuals with cancer."

Reviewer 2 Report

Comments and Suggestions for Authors

The narrative review by Adeline Fontvieille, Hugo Parent-Roberge , Tamás Fülöp, Michel Pavic and Eléonor Riesco entitled „The mechanisms underlying the beneficial impact of aerobic training on cancer-related fatigue: a conceptual review” summarizes the current state of knowledge on the benefits of physical activity to combat cancer-related fatigue.

After introducing cancer-related fatigue and the benefits of physical exercise, the narrative runs through the impact of exercise on (i) peripheral & neural inflammation, (ii) immune cells (NK, CD4+, CD8+) and (ii) alterations of the neuroendocrine system (cortico-tropin-releasing hormone, indoleamine-2,3-dioxygenase, adrenocorticotropic hormone). It ends in a summary model bringing the different elements together.

The text reads well and the story line works. The review is topical and of interest to a wider readership.

While I have no major comments, there are a few minor remarks:

L43 multidimensional syndrome might be better than construct

L 50 please make here a reference to Figure 1 as well

L70 Figure 1 shows the kinetics of CFR

L75 effect of anti-cancer treatment is less

L78 over successive intervals (effect of chronic exercise)

L90 a literature review precedes the content section to facilitate understanding

L99 by the chemotherapy, the immune system

L163 cancer patients to CRF.

L164 Natural Killer Cells (NK)

L203 this sentence is unclear “In fact, a study conducted by Bower et al. [63] reported a flattened diurnal cortisol slope, and a slow decline of cortisol in the evening, which argues dysfunction of the HPA axis in patients with CRF.” Please explain for the benefit of the reader

L209 Interestingly, indoleamine-2,3-dioxygenase…

L227 In healthy humans

L269-276 Figure 2A not (a) The same for the remaining panel letters in this section

Figure 2 would benefit from a brief figure legend giving the reader the key points of the summary figure. Each display item should be a stand-alone piece of information independently of the main text.

Comments on the Quality of English Language

Only minor editing is required

Author Response

Comments 1: The narrative review by Adeline Fontvieille, Hugo Parent-Roberge , Tamás Fülöp, Michel Pavic and Eléonor Riesco entitled „The mechanisms underlying the beneficial impact of aerobic training on cancer-related fatigue: a conceptual review” summarizes the current state of knowledge on the benefits of physical activity to combat cancer-related fatigue.

After introducing cancer-related fatigue and the benefits of physical exercise, the narrative runs through the impact of exercise on (i) peripheral & neural inflammation, (ii) immune cells (NK, CD4+, CD8+) and (ii) alterations of the neuroendocrine system (cortico-tropin-releasing hormone, indoleamine-2,3-dioxygenase, adrenocorticotropic hormone). It ends in a summary model bringing the different elements together.

The text reads well and the story line works. The review is topical and of interest to a wider readership.

Response 1: We would like to thank the reviewer for the general comments on the paper, and we also hope that this paper will be useful more widely in the field of oncology.

Comments 2: While I have no major comments, there are a few minor remarks:

-L43 multidimensional syndrome might be better than construct

-L 50 please make here a reference to Figure 1 as well

-L70 Figure 1 shows the kinetics of CFR

-L75 effect of anti-cancer treatment is less

-L78 over successive intervals (effect of chronic exercise)

-L90 a literature review precedes the content section to facilitate understanding

-L99 by the chemotherapy, the immune system

-L163 cancer patients to CRF.

-L164 Natural Killer Cells (NK)

-L203 this sentence is unclear “In fact, a study conducted by Bower et al. [63] reported a flattened diurnal cortisol slope, and a slow decline of cortisol in the evening, which argues dysfunction of the HPA axis in patients with CRF.” Please explain for the benefit of the reader

-L209 Interestingly, indoleamine-2,3-dioxygenase…

-L227 In healthy humans

Response 2: We agree with the above propositions, and we have accordingly modified these points. You can now find in the text this modification as follows (and in the manuscript with track changes).

-Page 1, Line 44: ” While recognized as a multidimensional syndrome encompassing emotional […]”.

-Page 2, Line 52: “[…] before treatment around 10 days post-chemotherapy (as shown in Figure 1).”.

-Page 2, Line 77, in the caption figure: “Figure 1 shows the kinetics […]”.

-Page 2, Line 85, in the caption figure: “and the effect of cumulative anti-cancer treatment is less intense compared to […].”.

-Page 2, Line 87, in the caption figure: “[…] might contribute to a gradual attenuation of CRF over time, potentially reflecting the cumulative impact of acute exercise.”.

-Page 3, Line 104: “[…] potential mechanisms explaining CRF precede the content section to facilitate understanding.”.

-Page 3, Line 113: “ In response to this acute disturbance in the body’s homeostatic, […]”.

-Page 4, Line 197: “ […] no evidence linking T-cell dysfunction in cancer patients to CRF.”.

-Page 4, Line 198: It appears you suggested adding the abbreviation of NK cells, but we have defined this earlier in the section at line 187, so we decided to make no changes regarding this issue.

-Page 5, Line 253: Thank you for your advice in the lack of clarity. We can now read “Supporting this notion, a study conducted by Bower et al. [70] reported higher daytime cortisol levels with a blunted circadian rhythm (flatter slope) compared to normal peak times (morning and evening), suggesting dysfunction of the HPA axis in patients with CRF.

-Page 5, Line 262: “Interestingly, indoleamine-2,3-dioxygenase (IDO) […]”.

-Page 5, Line 284: “In healthy humans, acute response to […]”.

Comments 3: L269-276 Figure 2A not (a) The same for the remaining panel letters in this section.

Figure 2 would benefit from a brief figure legend giving the reader the key points of the summary figure. Each display item should be a stand-alone piece of information independently of the main text.

Response 3: Your remarks about the Figure 2 and for the figure legend pointed out that the text following both figures (1 and 2) was not properly formatted, suggesting that the information was part of the main text. We made the correction and change for the “caption figure” formatting. Thank you for raising this issue.

5. Additional clarifications

Regarding the last comment of Reviewer 2, we would like to draw your attention to the fact that the text following the two figures should be read separately from the main text, as these are the captions. Additionally, we would like to acknowledge that both figures were created by the first author with Biorender.com and have not been previously published or reprinted. To our knowledge, there are no copyright issues.
